# FasterSeg: Searching for Faster Real-time Semantic Segmentation

**Wuyang Chen**[1*], **Xinyu Gong**[1*]**Xianming Liu**[2], **Qian Zhang**[2], **Yuan Li**[2], **Zhangyang Wang**[1]
[1]Department of Computer Science and Engineering, Texas A&M University
[2]Horizon Robotics Inc.
{wuyang.chen,xy_gong,atlaswang}@tamu.edu
{xianming.liu,qian01.zhang,yuan.li}@horizon.ai
https://github.com/TAMU-VITA/FasterSeg

## Abstract

We present **FasterSeg**, an automatically designed semantic segmentation network with not only state-of-the-art performance but also faster speed than current methods. Utilizing neural architecture search (NAS), FasterSeg is discovered from a novel and broader search space integrating multi-resolution branches, that has been recently found to be vital in manually designed segmentation models. To better calibrate the balance between the goals of high accuracy and low latency, we propose a decoupled and fine-grained latency regularization, that effectively overcomes our observed phenomenons that the searched networks are prone to "collapsing" to low-latency yet poor-accuracy models. Moreover, we seamlessly extend FasterSeg to a new collaborative search (co-searching) framework, simultaneously searching for a teacher and a student network in the same single run. The teacher-student distillation further boosts the student model's accuracy. Experiments on popular segmentation benchmarks demonstrate the competency of FasterSeg. For example, FasterSeg can run over 30% faster than the closest manually designed competitor on Cityscapes, while maintaining comparable accuracy.

## 1 Introduction

Semantic segmentation predicts pixel-level annotations of different semantic categories for an image. Despite its performance breakthrough thanks to the prosperity of convolutional neural networks (CNNs) (Long et al., 2015), as a dense structured prediction task, segmentation models commonly suffer from heavy memory costs and latency, often due to stacking convolutions and aggregating multiple-scale features, as well as the increasing input image resolutions. However, recent years witness the fast-growing demand for real-time usage of semantic segmentation, *e.g.*, autonomous driving. Such has motivated the enthusiasm on designing low-latency, more efficient segmentation networks, without sacrificing accuracy notably (Zhao et al., 2018; Yu et al., 2018a).

The recent success of neural architecture search (NAS) algorithms has shed light on the new horizon in designing better semantic segmentation models, especially under latency of other resource constraints. Auto-DeepLab (Liu et al., 2019a) first introduced network-level search space to optimize resolutions (in addition to cell structure) for segmentation tasks. Zhang et al. (2019) and Li et al. (2019) adopted pre-defined network-level patterns of spatial resolution, and searched for operators and decoders with latency constraint. Despite a handful of preliminary successes, we observe that the successful human domain expertise in designing segmentation models appears to be not fully integrated into NAS frameworks yet. For example, human-designed architectures for real-time segmentation (Zhao et al., 2018; Yu et al., 2018a) commonly exploit multi-resolution branches with proper depth, width, operators, and downsample rates, and find them contributing vitally to the success: such flexibility has not been unleashed by existing NAS segmentation efforts. Furthermore, the trade-off between two (somewhat conflicting) goals, *i.e.*, high accuracy and low latency, also makes the search process unstable and prone to "bad local minima" architecture options.

---

[*]Work done during the first two authors' research internships with Horizon Robotics Applied AI Lab.

As the well-said quote goes: *"those who do not learn history are doomed to repeat it"*. Inheriting and inspired by the successful practice in hand-crafted efficient segmentation, we propose a novel NAS framework dubbed **FasterSeg**, aiming to achieve extremely fast inference speed and competitive accuracy. We designed a special search space capable of supporting optimization over multiple branches of different resolutions, instead of a single backbone. These searched branches are adaptively aggregated for the final prediction. To further balance between accuracy versus latency and avoiding collapsing towards either metric (*e.g.*, good latency yet poor accuracy), we design a decoupled and fine-grained latency regularization, that facilitates a more flexible and effective calibration between latency and accuracy. Moreover, our NAS framework can be easily extended to a collaborative search (co-searching), *i.e.*, jointly searching for a complex teacher network and a light-weight student network in a single run, whereas the two models are coupled by feature distillation in order to boost the student's accuracy. We summarize our main contributions as follows:

- A novel NAS search space tailored for real-time segmentation, where multi-resolution branches can be flexibility searched and aggregated.
- A novel decoupled and fine-grained latency regularization, that successfully alleviates the "architecture collapse" problem in the latency-constrained search.
- A novel extension to teacher-student co-searching for the first time, where we distill the teacher to the student for further accuracy boost of the latter.
- Extensive experiments demonstrating that FasterSeg achieves extremely fast speed (over 30% faster than the closest manually designed competitor on CityScapes) and maintains competitive accuracy.

## 2  RELATED WORK

Human-designed CNN architectures achieve good accuracy performance nowadays (He et al., 2016; Chen et al., 2018b; Wang et al., 2019; Sun et al., 2019). However, designing architectures to balance between accuracy and other resource constraints (latency, memory, FLOPs, *etc.*) requires more human efforts. To free human experts from this challenging trade-off, neural architecture search (NAS) has been recently introduced and drawn a booming interest (Zoph & Le, 2016; Brock et al., 2017; Pham et al., 2018; Liu et al., 2018a; Chen et al., 2018a; Bender et al., 2018; Chen et al., 2018c; Gong et al., 2019). These works optimize both accuracy and resource utilization, via a combined loss function (Wu et al., 2019), or a hybrid reward signal for policy learning (Tan et al., 2019; Cheng et al., 2018), or a constrained optimization formulation (Dai et al., 2019).

Most existing resource-aware NAS efforts focus on classification tasks, while semantic segmentation has higher requirements for preserving details and rich contexts, therefore posing more dilemmas for efficient network design. Fortunately, previous handcrafted architectures for real-time segmentation have identified several consistent and successful design patterns. ENet (Paszke et al., 2016) adopted early downsampling, and ICNet (Zhao et al., 2018) further incorporated feature maps from multi-resolution branches under label guidance. BiSeNet (Yu et al., 2018a) fused a context path with fast downsampling and a spatial path with smaller filter strides. More works target on segmentation efficiency in terms of computation cost (He et al., 2019; Marin et al., 2019) and memory usage (Chen et al., 2019). Their multi-resolution branching and aggregation designs ensure **sufficiently large receptive fields** (contexts) while preserving **high-resolution fine details**, providing important clues on how to further optimize the architecture.

There have been recent studies that start pointing NAS algorithms to segmentation tasks. Auto-DeepLab (Liu et al., 2019a) pioneered in this direction by searching the cells and the network-level downsample rates, to flexibly control the spatial resolution changes throughout the network. Zhang et al. (2019) and Li et al. (2019) introduced resource constraints into NAS segmentation. A multi-scale decoder was also automatically searched (Zhang et al., 2019). However, compared with manually designed architectures, those search models still follow a single-backbone design and did not fully utilize the prior wisdom (*e.g.*, multi-resolution branches) in designing their search spaces.

Lastly, we briefly review knowledge distillation (Hinton et al., 2015), that aims to transfer learned knowledge from a sophisticated teacher network to a light-weight student, to improve the (more efficient) student's accuracy. For segmentation, Liu et al. (2019b) and Nekrasov et al. (2019) proposed to leverage knowledge distillation to improve the accuracy of the compact model and speed-up convergence. There was no prior work in linking distillation with NAS yet, and we will introduce the extension of FasterSeg by integrating teacher-student model collaborative search for the first time.

## 3 FASTERSEG: FASTER REAL-TIME SEGMENTATION

Our FasterSeg is discovered from an efficient and multi-resolution search space inspired by previous manual design successes. A fine-grained latency regularization is proposed to overcome the challenge of "architecture collapse" (Cheng et al., 2018). We then extend our FasterSeg to a teacher-student co-searching framework, further resulting in a lighter yet more accurate student network.

### 3.1 EFFICIENT SEARCH SPACE WITH MULTI-RESOLUTION BRANCHING

The core motivation behind our search space is to search multi-resolution branches with overall low latency, which has shown effective in previous manual design works (Zhao et al., 2018; Yu et al., 2018a). Our NAS framework automatically selects and aggregates branches of different resolutions, based on efficient cells with searchable superkernels.

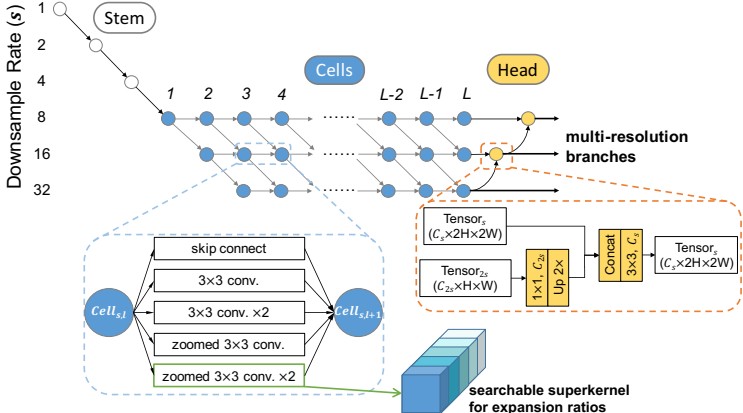

**Figure 1:** The multi-resolution branching search space for FasterSeg, where we aim to optimize multiple branches with different output resolutions. These outputs are progressively aggregated together in the head module. Each cell is individually searchable and may have two inputs and two outputs, both of different downsampling rates ($s$). Inside each cell, we enable searching for expansion ratios within a single superkernel.

#### 3.1.1 SEARCHABLE MULTI-RESOLUTION BRANCHES

Inspired by (Liu et al., 2019a), we enable searching for spatial resolutions within the $L$-layer cells (Figure 1), where each cell takes inputs from two connected predecessors and outputs two feature maps of different resolutions. Hand-crafted networks for real-time segmentation found multi-branches of different resolutions to be effective (Zhao et al., 2018; Yu et al., 2018a). However, architectures explored by current NAS algorithms are restricted to a single backbone.

Our goal is to select $b$ ($b > 1$) branches of different resolutions in this $L$-layer framework. Specifically, we could choose $b$ different final output resolutions for the last layer of cells, and decode each branch via backtrace (section 3.4). This enables our NAS framework to explore $b$ individual branches with different resolutions, which are progressively "learned to be aggregated" by the head module (Figure 1).

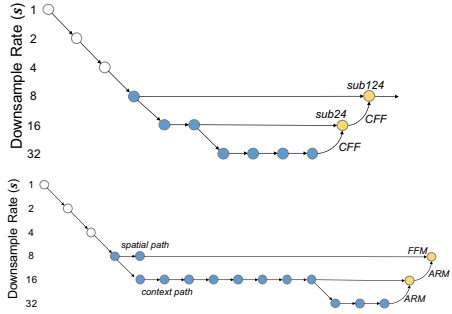

**Figure 2:** Our multi-resolution search space covers existing manual designs for real-time segmentation (unused cells omitted). Top: IC-Net (Zhao et al., 2018). Bottom: BiSeNet (Yu et al., 2018a)

We follow the convention to increase the number of channels at each time of resolution downsampling. To enlarge the model capacity without incurring much latency, we first downsample the input image to $\frac{1}{8}$ original scale with our stem module, and then set our searchable downsample rates $s \in \{8, 16, 32\}$. Figure 2 shows that our multi-resolution search space is able to cover existing human-designed networks for real-time segmentation. See Appendix B for branch selection details.

#### 3.1.2 CHOOSING EFFICIENT OPERATORS WITH LARGE RECEPTIVE FIELDS

As we aim to boost the inference latency, the speed of executing an operator is a direct metric (rather than indirect metrics like FLOPs) for selecting operator candidates $\mathcal{O}$. Meanwhile, as we previously

discussed, it is also important to ensure sufficiently large receptive field for spatial contexts. We analyze typical operators, including their common surrogate latency measures (FLOPs, parameter numbers), and their real-measured latency on an NVIDIA 1080Ti GPU with TensorRT library, and their receptive fields, as summarized in Table 1.

Compared with standard convolution, group convolution is often used for reducing FLOPs and number of parameters (Sandler et al., 2018; Ma et al., 2018). Convolving with two groups has the same receptive field with a standard convolution but is 13% faster, while halving the parameter amount (which might not be preferable as it reduces the model learning capacity). Dilated convolution has an enlarged receptive field and is popular in dense predictions (Chen et al., 2018b; Dai et al., 2017). However, as shown in Table 1 (and as widely acknowledged in engineering practice), dilated convolution (with dilation rate 2) suffers from dramatically higher latency, although that was not directly reflected in FLOPs nor parameter numbers. In view of that, we design a new variant called "zoomed convolution", where the input feature map is sequentially processed with bilinear downsampling, standard convolution, and bilinear upsampling. This special design enjoys 40% lower latency and 2 times larger receptive field compared to standard convolution. Our search space hence consists of the following operators:

- skip connection
- 3×3 conv.
- 3×3 conv. ×2
- "zoomed conv.": bilinear downsampling + 3×3 conv. + bilinear upsampling
- "zoomed conv. ×2": bilinear downsampling + 3×3 conv. ×2 + bilinear upsampling

As mentioned by Ma et al. (2018), network fragmentation can significantly hamper the degree of parallelism, and therefore practical efficiency. Therefore, we choose a sequential search space (rather than a directed acyclic graph of nodes (Liu et al., 2018b)), *i.e.*, convolutional layers are sequentially

**Table 1:** Specifications of different convolutions. Latency is measured using an input of size $1 \times 256 \times 32 \times 64$ on 1080Ti with TensorRT library. Each operator only contains one convolutional layer. The receptive field (RF) is relatively compared with the standard convolution (first row).

| Layer Type | Latency (ms) | FLOPs (G) | Params (M) | RF* |
|---|---|---|---|---|
| conv. | 0.15 | 1.21 | 0.59 | 1 |
| conv. group2 | 0.13 (−13%) | 0.60 (−50%) | 0.29 (−51%) | 1 |
| conv. dilation2 | 0.25 (+67%) | 1.10 (−9%) | 0.59 | 2 |
| zoomed conv. | 0.09 (−40%) | 0.30 (−75%) | 0.59 | 2 |

stacked in our network. In Figure 1, each cell is differentiable (Liu et al., 2018b; 2019a) and will contain only one operator, once the discrete architecture is derived (section 3.4). It is worth noting that we allow each cell to be individually searchable across the whole search space.

### 3.1.3 SEARCHABLE SUPERKERNEL FOR EXPANSION RATIOS

We further give each cell the flexibility to choose different channel expansion ratios. In our work, we search for the width of the connection between successive cells. That is however non-trivial due to the exponentially possible combinations of operators and widths. To tackle this problem, we propose a differentiably searchable superkernel, *i.e.*, directly searching for the expansion ratio $\chi$ within a single convolutional kernel which supports a set of ratios $\mathcal{X} \subseteq N^+$. Inspired by (Yu et al., 2018c) and (Stamoulis et al., 2019), from slim to wide our connections incrementally take larger subsets of input/output dimensions from the superkernel. During the architecture search, for each superkernel, only one expansion ratio is sampled, activated, and back-propagated in each step of stochastic gradient descent. This design contributes to a simplified and memory-efficient super network and is implemented via the renowned "Gumbel-Softmax" trick (see Appendix C for details).

To follow the convention to increase the number of channels as resolution downsampling, in our search space we consider the width $= \chi \times s$, where $s \in \{8, 16, 32\}$. We allow connections between each pair of successive cells flexibly choose its own expansion ratio, instead of using a unified single expansion ratio across the whole search space.

### 3.1.4 CONTINUOUS RELAXATION OF SEARCH SPACE

Denote the downsample rate as $s$ and layer index as $l$. To facilitate the search of spatial resolutions, we connect each cell with two possible predecessors' outputs with different downsample rates:

$$\overline{I}_{s,l} = \beta_{s,l}^0 \overline{O}_{\frac{s}{2} \to s, l-1} + \beta_{s,l}^1 \overline{O}_{s \to s, l-1} \tag{1}$$

Each cell could have at most two outputs with different downsample rates into its successors:

$$\overline{O}_{s \to s,l} = \sum_{k=1}^{|\mathcal{O}|} \alpha_{s,l}^k O_{s \to s,l}^k (\overline{I}_{s,l}, \chi_{s,l}^j, \text{stride} = 1)$$

$$\overline{O}_{s \to 2s,l} = \sum_{k=1}^{|\mathcal{O}|} \alpha_{s,l}^k O_{s \to 2s,l}^k (\overline{I}_{s,l}, \chi_{s,l}^j, \text{stride} = 2). \tag{2}$$

The expansion ratio $\chi_{s,l}^j$ is sampled via "Gumbel-Softmax" trick according to $p(\chi = \chi_{s,l}^j) = \gamma_{s,l}^j$. Here, $\alpha$, $\beta$, and $\gamma$ are all normalized scalars, associated with each operator $O^k \in \mathcal{O}$, each predecessor's output $\overline{O}_{l-1}$, and each expansion ratio $\chi \in \mathcal{X}$, respectively (Appendix D). They encode the architectures to be optimized and derived.

## 3.2 REGULARIZED LATENCY OPTIMIZATION WITH FINER GRANULARITY

Low latency is desirable yet challenging to optimize. Previous works (Cheng et al., 2018; Zhang et al., 2019) observed that during the search procedure, the supernet or search policy often fall into bad "local minimums" where the generated architectures are of extremely low latency but with poor accuracy, especially in the early stage of exploration. In addition, the searched networked tend to use more skip connections instead of choosing low expansion ratios (Shaw et al., 2019). This problem is termed as *"architecture collapse"* in our paper. The potential reason is that, finding architectures with extremely low latency (*e.g.* trivially selecting the most light-weight operators) is significantly easier than discovering meaningful compact architectures of high accuracy.

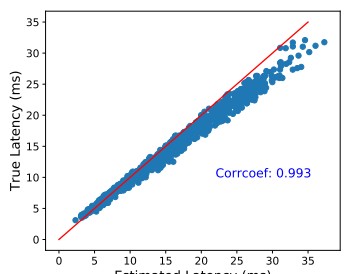

**Figure 3:** Correlation between network latency and its estimation via our latency lookup table (linear coefficient: 0.993). Red line indicates "$y = x$".

To address this "architecture collapse" problem, we for the first time propose to leverage a fine-grained, decoupled latency regularization. We first achieve the continuous relaxation of latency similar to the cell operations in section 3.1.4, via replacing the operator $O$ in Eqn. 1 and 2 with the corresponding latency. We build a latency lookup table that covers all possible operators to support the estimation of the relaxed latency. Figure 3 demonstrates the high correlation of 0.993 between the real and estimated latencies (see details in appendix E).

We argue that the core reason behind the "architecture collapse" problem is the different sensitivities of supernet to operator $O$, downsample rate $s$, and expansion ratio $\chi$. Operators like "3×3 conv. ×2" and "zoomed conv." have a huge gap in latency. Similar latency gap (though more moderate) exists between slim and wide expansion ratios. However, downsample rates like "8" and "32" do not differ much, since resolution downsampling also brings doubling of the number of both input and output channels.

**Table 2:** Supernet's sensitivity to latency under different granularities. Input size: (1, 3, 1024, 2048).

| Granularity | $\Delta$Latency (ms) |
|---|---|
| $O$ (operator) | 10.42 |
| $s$ (downsample rate) | 0.01 |
| $\chi$ (expansion ratio) | 5.54 |

We quantitatively compared the influence of $O$, $s$, and $\chi$ towards the supernet latency, by adjusting one of the three aspects and fixing the other two. Taking $O$ as the example, we first uniformly initialize $\beta$ and $\gamma$, and calculate $\Delta$Latency($O$) as the gap between the supernet which dominantly takes the slowest operators and the one adopts the fastest. Similar calculations were performed for $s$ and $\chi$. Values of $\Delta$Latency in Table 2 indicate the high sensitivity of the supernet's latency to operators and expansion ratios, while not to resolutions. Figure 4(a) shows that the unregularized latency optimization will bias the supernet towards light-weight operators and slim expansion ratios to quickly minimize the latency, ending up with problematic architectures with low accuracy.

Based on this observation, we propose a regularized latency optimization leveraging different granularities of our search space. We decouple the calculation of supernet's latency into three granularities of our search space ($O, s, \chi$), and regularize each aspect with a different factor:

$$\text{Latency}(O, s, \chi) = w_1 \text{Latency}(O|s, \chi) + w_2 \text{Latency}(s|O, \chi) + w_3 \text{Latency}(\chi|O, s) \tag{3}$$

where we by default set $w_1 = 0.001$, $w_2 = 0.997$, $w_3 = 0.002$[1]. This decoupled and fine-grained regularization successfully addresses this "architecture collapse" problem, as shown in Figure 4(b).

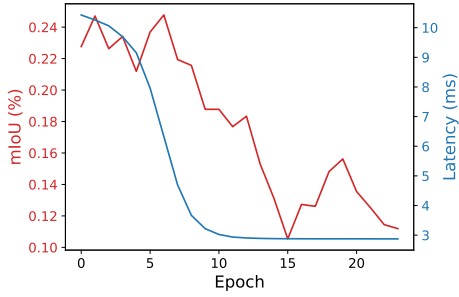 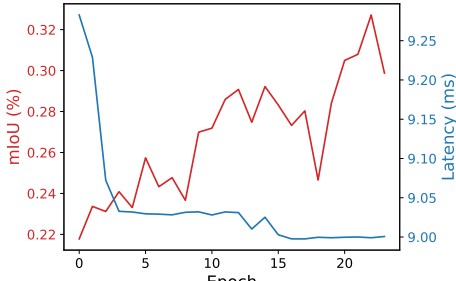

**(a)** Naive latency optimization.        **(b)** Proposed fine-grained latency regularization.

**Figure 4:** Comparing mIoU (%) and latency (ms) between supernets. The search is conducted on the Cityscapes training set and mIoU is measured on the validation set.

### 3.3 TEACHER/STUDENT CO-SEARCHING FOR KNOWLEDGE DISTILLATION

Knowledge Distillation is an effective approach to transfer the knowledge learned by a large and complex network (teacher $\mathcal{T}$) into a much smaller network (student $\mathcal{S}$). In our NAS framework, we can seamlessly extend to teacher-student co-searching, *i.e.*, collaboratively searching for two networks in a single run (Figure 5). Specifically, we search a complex teacher and light-weight student simultaneously via adopting **two sets of architectures in one supernet**: $(\alpha_{\mathcal{T}}, \beta_{\mathcal{T}})$ and $(\alpha_{\mathcal{S}}, \beta_{\mathcal{S}}, \gamma_{\mathcal{S}})$. Note that the teacher does not search the expansion ratios and always select the widest one.

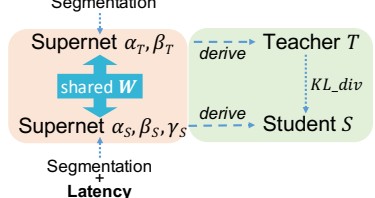

**Figure 5:** Our co-searching framework, which optimizes two architectures during search (left orange) and distills from a complex teacher to a light student during training from scratch (right green).

This extension does not bring any overhead in memory usage or size of supernet since the teacher and student share the same supernet weights $W$ during the search process. Two sets of architectures are iteratively optimized during search (please see details in Appendix F), and we apply the latency constraint only on the student, not on the teacher. Therefore, our searched teacher is a sophisticated network based on the same search space and supernet weights $W$ used by the student . During training from scratch, we apply a distillation loss from teacher $\mathcal{T}$ to student $\mathcal{S}$:

$$\ell_{\text{distillation}} = \mathbb{E}_{i \in \mathcal{R}} \text{KL}(\mathbf{q}_i^s || \mathbf{q}_i^t). \tag{4}$$

KL denotes the KL divergence. $\mathbf{q}_i^s$ and $\mathbf{q}_i^t$ are predicted logit for pixel $i$ from $\mathcal{S}$ and $\mathcal{T}$, respectively. Equal weights (1.0) are assigned to the segmentation loss and this distillation loss.

### 3.4 DERIVING DISCRETE ARCHITECTURES

Once the search is completed, we derive our discrete architecture from $\alpha$, $\beta$, and $\gamma$:
- $\alpha, \gamma$: We select the optimum operators and expansion ratios by taking the $\text{argmax}$ of $\alpha$ and $\gamma$. We shrink the operator "skip connection" to obtain a shallower architecture with less cells.
- $\beta$: Different from (Liu et al., 2019a), for each cell$_{s,l}$ we consider $\beta^0$ and $\beta^1$ as probabilities of two outputs from cell$_{\frac{s}{2},l-1}$ and cell$_{s,l-1}$ into cell$_{s,l}$. Therefore, by taking the $l^* = \text{argmax}_l(\beta_{s,l}^0)$, we find the optimum position (cell$_{s,l^*}$) where to downsample the current resolution ($\frac{s}{2} \to s$). [2]

It is worth noting that, the multi-resolution branches will share both cell weights and feature maps if their cells are of the same operator type, spatial resolution, and expansion ratio. This design contributes to a faster network. Once cells in branches diverge, the sharing between the branches will be stopped and they become individual branches (See Figure 6).

---

[1] These values are obtained by solving equations derived from Table 2 in order to achieve balanced sensitivities on different granularities: $10.42 \times w_1 = 0.01 \times w_2 = 5.54 \times w_1$, s.t. $w_1 + w_2 + w_3 = 1$.

[2] For a branch with two searchable downsampling positions, we consider the argmax over the joint probabilities $(l_1^*, l_2^*) = \text{argmax}_{l_1, l_2}(\beta_{s,l_1}^0 \cdot \beta_{2s,l_2}^0)$.

## 4 EXPERIMENTS

### 4.1 DATASETS AND IMPLEMENTATIONS

We use the Cityscapes (Cordts et al., 2016) as a testbed for both our architecture search and ablation studies. After that, we report our final accuracy and latency on Cityscapes, CamVid (Brostow et al., 2008), and BDD (Yu et al., 2018b). In all experiments, the class mIoU (mean Intersection over Union per class) and FPS (frame per second) are used as the metrics for accuracy and speed, respectively. Please see Appendix G for dataset details.

In all experiments, we use Nvidia Geforce GTX 1080Ti for benchmarking the computing power. We employ the high-performance inference framework TensorRT v5.1.5 and report the inference speed. During this inference measurement, an image of a batch size of 1 is first loaded into the graphics memory, then the model is warmed up to reach a steady speed, and finally, the inference time is measured by running the model for six seconds. All experiments are performed under CUDA 10.0 and CUDNN V7. Our framework is implemented with PyTorch. The search, training, and latency measurement codes are available at `https://github.com/TAMU-VITA/FasterSeg`.

### 4.2 ARCHITECTURE SEARCH

We consider a total of $L = 16$ layers in the supernet and our downsample rate $s \in \{8, 16, 32\}$. In our work we use number of branches $b = 2$ by default, since more branches will suffer from high latency. We consider expansion ratio $\chi_{s,l} \in \mathcal{X} = \{4, 6, 8, 10, 12\}$ for any "downsample rate" $s$ and layer $l$. The multi-resolution branches have 1695 unique paths. For cells and expansion ratios, we have $(1 + 4 \times 5)^{(15+14+13)} + 5^3 \approx 3.4 \times 10^{55}$ unique combinations. This results in a search space in the order of $10^{58}$, which is much larger and challenging, compared with preliminary studies.

Architecture search is conducted on Cityscapes training dataset. Figure 6 visualizes the best spatial resolution discovered (FasterSeg). Our FasterSeg achieved mutli-resolutions with proper depths. The two branches share the first three operators then diverge, and choose to aggregate outputs with downsample rates of 16 and 32. Oper-

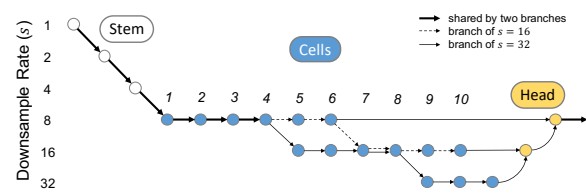

**Figure 6:** FasterSeg network discovered by our NAS framework.

ators and expansion ratios are listed in Table 7 in Appendix I, where the zoomed convolution is heavily used, suggesting the importance of low latency and large receptive field.

### 4.3 EVALUATION OF MULTI-RESOLUTION SEARCH SPACE AND CO-SEARCHING

We conduct ablation studies on Cityscapes to evaluate the effectiveness of our NAS framework. More specifically, we examine the impact of operators ($O$), downsample rate ($s$), expansion ratios ($\chi$), and also distillation on the accuracy and latency. When we expand from a single backbone ($b = 1$) to multi-branches ($b = 2$), our FPS drops but we gain a much improvement on mIoU, indicating the multi-resolution design is beneficial for

**Table 3:** Ablation studies of different search and training strategies. mIoU is measured on Cityscapes validation set. The input resolution is $1024 \times 2048$. $O$: operator; $s$: downsample rate; $\chi$: expansion ratios; $b$: number of branches; "$\rightarrow$": knowledge distillation.

| Settings | mIoU (%) | FPS | FLOPs | #Params |
|---|---|---|---|---|
| $O, s \mid \chi = 8, b = 1$ | 66.9 | 177.8 | 27.0 G | 6.3 M |
| $O, s \mid \chi = 8, b = 2$ | 69.5 | 119.9 | 42.8 G | 10.8 M |
| Teacher ($\mathcal{T}$) Student ($\mathcal{S}$) Co-searching | | | | |
| $\mathcal{S}$: $O, s, \chi \mid b = 2$ | 70.5 | 163.9 | 28.2 G | 4.4 M |
| $\mathcal{T} \rightarrow$ pruned $\mathcal{T}$ | 66.1 | 146.7 | 29.5 G | 4.7 M |
| $\mathcal{T} \rightarrow \mathcal{S}$(**FasterSeg**) | **73.1** | **163.9** | 28.2 G | 4.4 M |

segmentation task. By enabling the search for expansion ratios ($\chi$), we discover a faster network with FPS 163.9 without sacrificing accuracy (70.5%), which proves that the searchable superkernel gets the benefit from eliminating redundant channels while maintaining high accuracy. This is our student network ($\mathcal{S}$) discovered in our co-searching framework (see below).

We further evaluate the efficacy of our teacher-student co-searching framework. After the collaboratively searching, we obtain a teacher architecture ($\mathcal{T}$) and a student architecture ($\mathcal{S}$). As mentioned above, $\mathcal{S}$ is searched with searchable expansion ratios ($\chi$), achieving an FPS of 163.9 and an mIoU of 70.5%. In contrast, when we directly compress the teacher (channel pruning via selecting the

slimmest expansion ratio) and train with distillation from the well-trained original cumbersome teacher, it only achieved mIoU = 66.1% with only FPS = 146.7, indicating that our architecture co-searching surpass the pruning based compression. Finally, when we adopt the knowledge distillation from the well-trained cumbersome teacher to our searched student, we boost the student's accuracy to 73.1%, which is our final network **FasterSeg**. This demonstrates that both a student discovered by co-searching and training with knowledge distillation from the teacher are vital for obtaining an accurate faster real-time segmentation model.

### 4.4 REAL-TIME SEMANTIC SEGMENTATION

In this section, we compare our FasterSeg with other works for real-time semantic segmentation on three popular scene segmentation datasets. Note that since we target on real-time segmentation, we measure the mIoU without any evaluation tricks like flipping, multi-scale, etc.

**Cityscapes:** We evaluate FasterSeg on Cityscapes validation and test sets. We use original image resolution of 1024×2048 to measure both mIoU and speed inference. In Table 4, we see the superior FPS (163.9) of our FasterSeg, even under the maximum image resolution. This high FPS is over 1.3× faster than human-designed

**Table 4:** mIoU and inference FPS on Ciytscapes validation and test sets.

| Method | mIoU (%) | | FPS | Resolution |
|--------|----------|------|-----|------------|
| | val | test | | |
| ENet (Paszke et al., 2016) | - | 58.3 | 76.9 | 512×1024 |
| ICNet (Zhao et al., 2018) | 67.7 | 69.5 | 37.7 | 1024×2048 |
| BiSeNet (Yu et al., 2018a) | 69.0 | 68.4 | 105.8 | 768×1536 |
| CAS (Zhang et al., 2019) | 71.6 | 70.5 | 108.0 | 768×1536 |
| Fast-SCNN (Poudel et al., 2019) | 68.6 | 68.0 | 123.5 | 1024×2048 |
| DF1-Seg-d8 (Li et al., 2019) | 72.4 | 71.4 | 136.9 | 1024×2048 |
| FasterSeg (ours) | **73.1** | **71.5** | **163.9** | 1024×2048 |

networks. Meanwhile, our FasterSeg still maintains competitive accuracy, which is 73.1% on the validation set and 71.5% on the test set. This accuracy is achieved with only Cityscapes fine-annotated images, without using any extra data (coarse-annotated images, ImageNet, etc.).

**CamVid:** We directly transfer the searched architecture on Cityscapes to train on CamVid. Table 5 reveals that without sacrificing much accuracy, our FasterSeg achieved an FPS of 398.1. This extremely high speed is over 47% faster than the closest competitor in FPS (Yu et al., 2018a), and is over two times faster than the work with the best mIoU (Zhang et al., 2019). This impressive result verifies both the high performance of FasterSeg and also the transferability of our NAS framework.

**BDD:** In addition, we also directly transfer the learned architecture to the BDD dataset. In Table 6 we compare our FasterSeg with the baseline provided by Yu et al. (2018b). Since no previous work has considered real-time segmentation on the BDD dataset, we get 15 times faster than the DRN-D-22 with slightly higher mIoU. Our FasterSeg still preserve the extremely fast speed and competitive accuracy on BDD.

**Table 5:** mIoU and inference FPS on CamVid test set. The input resolution is 720 × 960.

| Method | mIoU (%) | FPS |
|--------|----------|-----|
| ENet (Paszke et al., 2016) | 68.3 | 61.2 |
| ICNet (Zhao et al., 2018) | 67.1 | 27.8 |
| BiSeNet (Yu et al., 2018a) | 65.6 | 269.1 |
| CAS (Zhang et al., 2019) | **71.2** | 169.0 |
| FasterSeg (ours) | 71.1 | **398.1** |

**Table 6:** mIoU and inference FPS on BDD validation set. The input resolution is 720 × 1280.

| Method | mIoU (%) | FPS |
|--------|----------|-----|
| DRN-D-22 (Yu et al., 2017) | 53.2 | 21.0 |
| DRN-D-38 (Yu et al., 2017) | **55.2** | 12.9 |
| FasterSeg (ours) | 55.1 | **318.0** |

## 5 CONCLUSION

We introduced a novel multi-resolution NAS framework, leveraging successful design patterns in handcrafted networks for real-time segmentation. Our NAS framework can automatically discover FasterSeg, which achieved both extremely fast inference speed and competitive accuracy. Our search space is intrinsically of low-latency and is much larger and challenging due to flexible searchable expansion ratios. More importantly, we successfully addressed the "architecture collapse" problem, by proposing the novel regularized latency optimization of fine-granularity. We also demonstrate that by seamlessly extending to teacher-student co-searching, our NAS framework can boost the student's accuracy via effective distillation.

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

## A    STEM AND HEAD MODULE

**Stem:** Our stem module aims to quickly downsample the input image to $\frac{1}{8}$ resolution while increasing the number of channels. The stem module consists of five $3 \times 3$ convolution layers, where the first, second, and fourth layer are of stride two and double the number of channels.

**Head:** As shown in Figure 1, feature map of shape $(C_{2s} \times H \times W)$ is first reduced in channels by a $1 \times 1$ convolution layer and bilinearly upsampled to match the shape of the other feature map $(C_s \times 2H \times 2W)$. Then, two feature maps are concatenated and fused together with a $3 \times 3$ convolution layer. Note that we not necessarily have $C_{2s} = 2C_s$ because of the searchable expansion ratios.

## B    BRANCH SELECTION CRITERION

Since our searchable downsample rates $s \in \{8, 16, 32\}$ and the number of selected branches $b = 2$, our supernet needs to select branches of three possible combinations of resolutions: $\{8, 16\}$, $\{8, 32\}$, and $\{16, 32\}$. For each combination, branches of two resolutions will be aggregated by our head module.

Our supernet selects the best $b$ branches based on the criterion used in (Tan et al., 2019):

$$Target(m) = ACC(m) \times [\frac{LAT(m)}{T}]^w, \tag{5}$$

where $m$ is a searched model aggregating $b$ branches, with accuracy $ACC(m)$ and latency $LAT(m)$. $w$ is the weight factor defined as:

$$w = \begin{cases} \alpha, & \text{if } LAT(m) \leq T \\ \beta, & \text{otherwise.} \end{cases} \tag{6}$$

We empirically set $\alpha = \beta = -0.07$ and the target latency $T = 8.3$ ms in our work.

## C    "GUMBEL-SOFTMAX" TRICK FOR SEARCHING EXPANSION RATIOS

Formally, suppose we have our set of expansion ratios $\mathcal{X} \subseteq N^+$, and we want to sample one ratio $\chi$ from $\mathcal{X}$. For each $\chi^i$ we have an associated probability $\gamma^i$, where $\sum_{i=1}^{|\mathcal{X}|} \gamma^i = 1$. "Gumbel-Softmax" trick (Gumbel, 1954; Maddison et al., 2014) helps us approximate differentiable sampling. We first sample a "Gumbel-Noise" $o^i = -\log(-\log(u))$ with $u \sim \text{Unif}[0, 1]$. Instead of selecting $\chi^j$ such that $j = \text{argmax}_i(\gamma^i)$, we choose $j = \text{argmax}_i\left(\frac{\exp(\frac{\log(\gamma^i)+o^i}{\tau})}{\sum_{m=1}^{|\mathcal{X}|} \exp(\frac{\log(\gamma^m)+o^m}{\tau})}\right)$. We set the temperature parameter $\tau = 1$ in our work.

## D    NORMALIZED SCALARS $\alpha, \beta, \gamma$

$\alpha$, $\beta$, and $\gamma$ are all normalized scalars and implemented as softmax. They act as probabilities associating with each operator $O^k \in \mathcal{O}$, each predecessor's output $\overline{O}_{l-1}$, and each expansion ratio $\chi \in \mathcal{X}$, respectively:

$$\sum_{k=1}^{|\mathcal{O}|} \alpha_{s,l}^k = 1, \forall s, l \quad \text{and} \quad \alpha_{s,l}^k \geq 0, \forall k, s, l$$

$$\beta_{s,l}^0 + \beta_{s,l}^1 = 1, \forall s, l \quad \text{and} \quad \beta_{s,l}^0, \beta_{s,l}^1 \geq 0, \forall s, l \tag{7}$$

$$\sum_{j=1}^{|\mathcal{X}|} \gamma_{s,l}^j = 1, \forall s, l \quad \text{and} \quad \gamma_{s,l}^j \geq 0, \forall j, s, l,$$

where $s$ is downsample rate and $l$ is index of the layer in our supernet.

## E    LATENCY ESTIMATION

We build a latency lookup table that covers all possible situations and use this lookup table as building blocks to estimate the relaxed latency. To verify the continuous relaxation of latency, we

randomly sample networks of different operators/downsample rates/expansion ratios out of the supernet $\mathcal{M}$, and measured both the real and estimated latency. We estimate the network latency by accumulating all latencies of operators consisted in the network. In Figure 3, we can see the high correlation between the two measurements, with a correlation coefficient of 0.993. This accurate estimation of network latency benefits from the sequential design of our search space.

## F TRAINING

Given our supernet $\mathcal{M}$, the overall optimization target (loss) during architecture search is:

$$L = L_{\text{seg}}(\mathcal{M}) + \lambda \cdot Lat(\mathcal{M}) \tag{8}$$

We adopt cross-entropy with "online-hard-element-mining" as our segmentation loss $L_{\text{seg}}$. $Lat(\mathcal{M})$ is the continuously relaxed latency of supernet, and $\lambda$ is the balancing factor. We set $\lambda = 0.01$ in our work.

As the architecture $\alpha, \beta$, and $\gamma$ are now involved in the differentiable computation graph, they can be optimized using gradient descent. Similar in (Liu et al., 2019a), we adopt the first-order approximation ((Liu et al., 2018b)), randomly split our training dataset into two disjoint sets trainA and trainB, and alternates the optimization between:

1. Update network weights $W$ by $\nabla_W L_{\text{seg}}(\mathcal{M}|W, \alpha, \beta, \gamma)$ on trainA,
2. Update architecture $\alpha, \beta, \gamma$ by $\nabla_{\alpha,\beta,\gamma} L_{\text{seg}}(\mathcal{M}|W, \alpha, \beta, \gamma) + \lambda \cdot \nabla_{\alpha,\beta,\gamma} LAT(\mathcal{M}|W, \alpha, \beta, \gamma)$ on trainB.

When we extend to our teacher-student co-searching where we have two sets of architectures $(\alpha_{\mathcal{T}}, \beta_{\mathcal{T}})$ and $(\alpha_{\mathcal{S}}, \beta_{\mathcal{S}}, \gamma_{\mathcal{S}})$, our alternative optimization becomes:

1. Update network weights $W$ by $\nabla_W L_{\text{seg}}(\mathcal{M}|W, \alpha_{\mathcal{T}}, \beta_{\mathcal{T}})$ on trainA,
2. Update network weights $W$ by $\nabla_W L_{\text{seg}}(\mathcal{M}|W, \alpha_{\mathcal{S}}, \beta_{\mathcal{S}}, \gamma_{\mathcal{S}})$ on trainA,
3. Update architecture $\alpha_{\mathcal{T}}, \beta_{\mathcal{T}}$ by $\nabla_{\alpha_{\mathcal{T}}, \beta_{\mathcal{T}}, \gamma_{\mathcal{T}}} L_{\text{seg}}(\mathcal{M}|W, \alpha_{\mathcal{T}}, \beta_{\mathcal{T}})$ on trainB.
4. Update architecture $\alpha_{\mathcal{S}}, \beta_{\mathcal{S}}, \gamma_{\mathcal{S}}$ by $\nabla_{\alpha_{\mathcal{S}}, \beta_{\mathcal{S}}, \gamma_{\mathcal{S}}} L_{\text{seg}}(\mathcal{M}|W, \alpha_{\mathcal{S}}, \beta_{\mathcal{S}}, \gamma_{\mathcal{S}}) + \lambda \cdot \nabla_{\alpha_{\mathcal{S}}, \beta_{\mathcal{S}}, \gamma_{\mathcal{S}}} LAT(\mathcal{M}|W, \alpha_{\mathcal{S}}, \beta_{\mathcal{S}}, \gamma_{\mathcal{S}})$ on trainB.

In all search experiments, we first pretrain the supernet without updating the architecture parameter for 20 epochs, then start architecture searching for 30 epochs.

**Optimizing Multiple Widths**: To train multiple widths within a superkernel, it is unrealistic to accumulate losses from all different width options in the super network. Therefore, we approximate the multi-widths optimization by training the maximum and minimum expansion ratios. During pretraining the super network, we train each operator with the smallest width, largest width, and 2 random widths. When we are searching for architecture parameters, we train each operator with the smallest width, largest width, and the width that is sampled from the current $\gamma$ by Gumbel Softmax.

## G BENCHMARK DATASETS

**Cityscapes** (Cordts et al., 2016) is a popular urban street scene dataset for semantic segmentation. It contains high-quality pixel-level annotations, 2,975 images for training and 500 images for validation. In our experiments, we only use the fine annotated images without any coarse annotated image or extra data like ImageNet (Deng et al., 2009). For testing, it offers 1,525 images without ground-truth for a fair comparison. These images all have a resolution of $1024 \times 2048$ (H×W), where each pixel is annotated to pre-defined 19 classes.

**CamVid** (Brostow et al., 2008) is another street scene dataset, extracted from five video sequences taken from a driving automobile. It contains 701 images in total, where 367 for training, 101 for validation and 233 for testing. The images have a resolution of $720 \times 960$ (H×W) and 11 semantic categories.

**BDD**: The BDD (Yu et al., 2018b) is a recently released urban scene dataset. For the segmentation task, it contains 7,000 images for training and 1,000 for validation. The images have a resolution of $720 \times 1280$ (H×W) and shared the same 19 semantic categories as used in the Cityscapes.

# H ARCHITECTURE SEARCH IMPLEMENTATIONS

As stated in the second line of Eqn. 2, a stride 2 convolution is used for all s → 2s connections, both to reduce spatial size and double the number of filters. Bilinear upsampling is used for all upsampling operations.

We conduct architecture search on the Cityscapes dataset. We use $160 \times 320$ random image crops from half-resolution ($512 \times 1024$) images in the training set. Note that the original validation set or test set is never used for our architecture search. When learning network weights $W$, we use SGD optimizer with momentum 0.9 and weight decay of $5\times10^{-4}$. We used the exponential learning rate decay of power 0.99. When learning the architecture parameters $\alpha, \beta, and \gamma$, we use Adam optimizer with learning rate $3\times10^{-4}$. The entire architecture search optimization takes about 2 days on one 1080Ti GPU.

# I FASTERSEG STRUCTURE

In Table 7 we list the operators ($O$) and expansion ratios ($\chi$) selected by our FasterSeg. The down-sample rates $s$ in Table 7 and Figure 6 match. We have the number of output channels $c_{out} = s \times \chi$. We observed that the zoomed convolution is heavily used, suggesting the importance of low latency and large receptive field.

**Table 7:** Cells used in FasterSeg. Left: cells for branch with final downsample rate of 16. Right: cells for branch with final downsample rate of 32. $s$: downsample rate. $\chi$: expansion ratio. c_out: number of output channels.

| Cell | Operator | $s$ | $\chi$ | c_out |
|---|---|---|---|---|
| 1 | conv. $\times2$ | 8 | 4 | 32 |
| 2 | conv. $\times2$ | 8 | 4 | 32 |
| 3 | zoomed conv. $\times2$ | 8 | 4 | 32 |
| 4 | zoomed conv. $\times2$ | 8 | 4 | 32 |
| 5 | conv. $\times2$ | 8 | 4 | 32 |
| 6 | conv. $\times2$ | 8 | 4 | 32 |
| 7 | zoomed conv. $\times2$ | 16 | 4 | 64 |
| 8 | zoomed conv. $\times2$ | 16 | 12 | 192 |
| 9 | zoomed conv. $\times2$ | 16 | 8 | 128 |

| Cell | Operator | $s$ | $\chi$ | c_out |
|---|---|---|---|---|
| 1 | conv. $\times2$ | 8 | 4 | 32 |
| 2 | conv. $\times2$ | 8 | 4 | 32 |
| 3 | zoomed conv. $\times2$ | 8 | 4 | 32 |
| 4 | zoomed conv. | 8 | 8 | 64 |
| 5 | zoomed conv. $\times2$ | 16 | 4 | 64 |
| 6 | zoomed conv. $\times2$ | 16 | 4 | 64 |
| 7 | zoomed conv. $\times2$ | 16 | 4 | 64 |
| 8 | zoomed conv. $\times2$ | 16 | 4 | 64 |
| 9 | zoomed conv. $\times2$ | 32 | 4 | 128 |
| 10 | zoomed conv. $\times2$ | 32 | 8 | 256 |

# J VISUALIZATION

We show some visualized segmentation results in Figure 7. From the third to the forth column ("$O, s | \chi = 8, b = 1$" to "$O, s | \chi = 8, b = 2$"), adding the extra branch of different scales provides more consistent segmentations (in the building and sidewalk area). From the fifth to the sixth column ("$\mathcal{T} \rightarrow$ pruned $\mathcal{T}$" to FasterSeg), our FasterSeg surpasses the pruned teacher network, where the segmentation is smoother and more accurate.

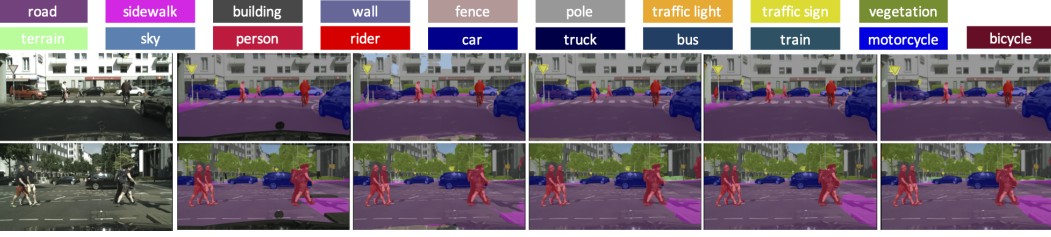

**Figure 7:** Visualization on Cityscapes validation set. Columns from left to right correspond to original images, ground truths, and results of "$O, s | \chi = 8, b = 1$", "$O, s | \chi = 8, b = 2$", "$\mathcal{T} \rightarrow$ pruned $\mathcal{T}$", FasterSeg (see Table 3 for annotation details). Best viewed in color.

