# OpenReview forum: "FasterSeg: Searching for Faster Real-time Semantic Segmentation"
_ICLR.cc/2020/Conference — Accept (Poster)_

### Official Review · AnonReviewer1 · 2019-10-22
**Official Blind Review #1**

**Rating:** 8

**Review:**

This paper presents an automatically designed semantic segmentation network utilising neural architecture search. The proposed method is discovered from a search space integrating multi-resolution branches, that has been recently found to be vital in manually designed segmentation models. To calibrate the balance between the goals of high accuracy and low latency, the authors propose a decoupled and fine-grained latency regularization, that effectively overcomes the observed phenomenons that the searched networks are prone to “collapsing” to low-latency yet poor-accuracy models. Moreover, the authors extend the proposed method to a new collaborative search (co-searching) framework, simultaneously searching for a teacher and a student network in the same single run. The teacher-student distillation further boosts the student model’s accuracy. Experimental results on Cityscapes, CamVid, and BDD verified the efficacy of the proposed method.

The writing is clear and the presentation is good. I like the motivation of this paper. The problem solved in this paper aligns with reality.

My concerns regarding this paper are as below.
1) The datasets used for evaluation are quite old except BDD, which make the results not so convincing. More experiments on more recent challenging semantic segmentation benchmarks are needed to verify the superiorities claimed in this paper. e.g., MHP v2.0 [Zhao et al., ACM MM 2018], etc.
2) The methods compared in Tab.6 are quite old, please add comparisons with more recent SOTAs.
3) Please add qualitative analysis to gain insight into the proposed method and to show why works better than other SOTAs.
4) The results in Tab.4,5,6 of the proposed method are not the best for mIoU on testing protocol, is this a trade-off between acc and speed?

Based on the above overall comments, I decide to give the rate of Weak Accept for this paper.

**Experience Assessment:**

I have published in this field for several years.

**Review Assessment: Checking Correctness Of Derivations And Theory:**

I carefully checked the derivations and theory.

**Review Assessment: Checking Correctness Of Experiments:**

I carefully checked the experiments.

**Review Assessment: Thoroughness In Paper Reading:**

I read the paper thoroughly.

---

> ### Author Response · Authors · 2019-11-11
> **Response to Review 1**
>
> Thank you for appreciating our novelty, worthy contribution, and strong results! We have addressed your remaining concerns below.
>
> Q1: The datasets used for evaluation are quite old except BDD, which make the results not so convincing. More experiments on more recent challenging semantic segmentation benchmarks are needed to verify the superiorities claimed in this paper. e.g., MHP v2.0 [Zhao et al., ACM MM 2018], etc.
> We are definitely interested in adopting our search algorithm to the MHP dataset (on human body parsing) in future. Currently, we follow the convention in segmentation works (“BiSeNet” Yu et al. 2018; “CAS” Zhang et al. 2019; “Partial Order Pruning” Li et al. 2019;) to choose popular scene segmentation datasets. Till latest works, all three datasets are still actively studied, and the Citycapes leaderboard is still actively being updated.
>
> Q2: The methods compared in Tab.6 are quite old, please add comparisons with more recent SOTAs.
> We thoroughly searched for public segmentation works on the BDD dataset and did not find more works related to our settings. As of the date we drafted the rebuttal, the BDD dataset paper has 173 citations. Most works on BDD consider domain adaptation settings. Relatively less has been done on BDD’s segmentation side (compared to Cityscapes), although it is increasingly considered as a challenging segmentation testbed too. Up to our best knowledge, we have chosen the latest and most competitive SOTAs in Table 6.
>
> Q3: Please add qualitative analysis to gain insight into the proposed method and to show why works better than other SOTAs.
> We have add visualizations in section J in appendix.  In Figure 7, from the third to the forth column, adding the extra branch of different scales provides more consistent segmentations (in the building and sidewalk area). From the fifth to the sixth column, our FasterSeg surpasses the prunned teacher network, where the segmentation is smoother and more accurate.
> The core reason our FasterSeg performs better than other peer works is combining merits from both NAS and human experts. Inspired by knowledge from previous works, multi-scale branches and large receptive fields play an important role for segmentation, and sequential (instead of complex and fragmented) blocks are more suitable for the low-latency purpose. These two priors motivate us to design a novel multi-scale sequential search space equipped with specialized light-weight operators (the “zoomed convolution”). Our regularized latency optimization achieves a better tradeoff between accuracy and latency than human-designed networks and avoids the architecture collapse problem found in other NAS works. Furthermore, our co-searched teacher network boosts the student’s accuracy.
>
> Q4: The results in Tab.4,5,6 of the proposed method are not the best for mIoU on testing protocol, is this a trade-off between acc and speed?
> In the experiments, by further fine-tuning our FasterSeg, we are now able to achieve 71.5% on Cityscapes test set, bypassing all previous works on both mIoU and FPS in Table 4, making CityScape our clear all-win case. In Table 5 and 6, our FasterSeg now achieves 71.1% and 54.9% respectively, reaching the same bar on mIoU as CAS and DRN, while surpassing their latency by over-doubling the FPS. Technically yes, we can adjust the tradeoff between accuracy and latency.

---

### Official Review · AnonReviewer3 · 2019-10-22
**Official Blind Review #3**

**Rating:** 8

**Review:**

Summary:
- key problem: neural architecture search (NAS) to improve both accuracy and runtime efficiency of deep nets for semantic segmentation;
- contributions: 1) a novel NAS search space leveraging multi-resolution branches, efficient operators ("zoomed convolutions"), and parametrized expansion ratios, 2) a decomposition and normalization of the latency objective to avoid a bias towards very fast but weak architectures, 3) a natural extension of the optimization problem to simultaneously search for teacher and student architectures in one pass, 4) a novel state-of-the-art efficient architecture (FasterSeg) found by the aforementioned NAS algorithm, 5) a detailed experimental evaluation on 3 datasets and an ablative analysis quantifying the benefits of the aforementioned contributions.

Recommendation: Accept.

Key reason 1: solid experimental results backing the claims.
- When compared to related efficient architectures, the proposed method results in competitive accuracy at significantly higher frame rates.
- This is validated on Cityscapes, CamVid, and BDD with the architecture found on Cityscapes.
- The resulting architecture (FasterSeg) is actually interpretable and makes sense, extending the handcrafted architectures used as inspiration.
- The ablative analysis shows that the numerous individual contributions are significant, esp. the multi-branch formulation and student co-searching.

Key reason 2: well-motivated method with a collection of multiple novel contributions that are interesting and practical.
- The multi-resolution branches formulation is simple and extends typical NAS focusing on single paths through the supernet.
- Teacher/student co-searching via learning two sets of architectures in one supernet seems novel, simple, and effective. Always picking the largest expansion ratios for the teacher and applying a distillation loss in addition to the latency loss for the student is sensible and seems to beat the standard pruning approach at no significant extra cost during NAS.
- The zoomed convolution operator seems like a novel efficient alternative to (expensive) dilated convolutions. Although it is very simple (bilinear downsampling -> 3x3 conv -> bilinear upsampling), it is not commonly used as an operator (as far as I know), and yet is found to be a key part of the final architecture (Table 7 appendix I) due to its low latency. The closest related operator / block I could think of might be blocks found in stacked hourglass networks (Newell et al).
- The optimization of the expansion ratios using the Gumbel-Softmax trick is interesting, although this is also explored in the very recent paper by Shaw et al. 2019 (possibly the closest related work that should be discussed in a bit more depth in Section 2);
- Decomposing and normalizing the latency objective to avoid "architecture collapse" (convergence to anemic architectures stemming from certain architectural factors dominating latency) is principled and effective.
- Caveat regarding novelty: I could not find the ideas proposed here in the literature, but its hard to be sure due to 1) the recent explosion of NAS papers, 2) the simplicity of certain ideas (e.g., "zoomed convolutions").


Additional Feedback:
- how is the student trained after NAS? Is the teacher first retrained from scratch? Is the student retrained from scratch on the teacher (after NAS or retraining)? in general, more details on what happens after co-searching would be helpful;
- "human designed CNN architectures achieve superior accuracy performance nowadays": this is a surprising statement considering the cited NAS papers report performance improvements (e.g., Zoph and Le 2016);
- missing reference also using multi-scale NAS for efficient and accurate semantic segmentation: "Searching for Efficient Multi-Scale Architectures for Dense Image Prediction", Chen et al, NeurIPS 2018;
- missing reference on NAS for efficient semantic segmentation that also uses distillation: "Fast neural architecture search of compact semantic segmentation models via auxiliary cells", Nekrasov et al, CVPR 2019;
- missing reference on joint NAS and quantization: "Joint Neural Architecture Search and Quantization", Chen et al, arxiv 2018;
- "we choose a sequential search space (rather than a directed acyclic graph of nodes (Liu et al., 2018b)), i.e., convolutional layers are sequentially stacked in our network": "stacked" is confusing here;
- "we allow each cell to be individually searchable across the whole search space": what do you mean? Anything beyond each cell containing different operators after learning?
- if \alpha = \beta in eq. 6 of appendix C, then w and hence Target(m) does not depend on latency, isn't this a typo?
- "Gumbel-Max" is typically called "Gumbel-Softmax" (cf. "Categorical Reparameterization with Gumbel-Softmax", Jang et al, ICLR'17);
- typos: "find them contribute", "the closet competitor", "is popular dense predictions".

**Experience Assessment:**

I have read many papers in this area.

**Review Assessment: Checking Correctness Of Derivations And Theory:**

I assessed the sensibility of the derivations and theory.

**Review Assessment: Checking Correctness Of Experiments:**

I carefully checked the experiments.

**Review Assessment: Thoroughness In Paper Reading:**

I read the paper thoroughly.

---

> ### Author Response · Authors · 2019-11-11
> **Response to Review 3**
>
> Thank you for appreciating our novelty, worthy contribution, and strong results! We have addressed your remaining concerns below.
>
> Q1: How is the student trained after NAS? Is the teacher first retrained from scratch? Is the student retrained from scratch on the teacher (after NAS or retraining)?
> The student and teacher are trained together from scratch, and the student is trained with both cross-entropy and distillation loss from the teacher. Since the teacher chooses heavy operators and expansion ratios, the distillation from the teacher to the student helps boost the accuracy (Table 3).
>
> Q2: "human designed CNN architectures achieve superior accuracy performance nowadays": this is a surprising statement considering the cited NAS papers report performance improvements (e.g., Zoph and Le 2016);
> Sorry for the wrong word. We actually just tried to say human designed CNNs are good in accuracy (while NAS can be better). We have revised this sentence in paper.
>
> Q3,4,5: Missing references.
> Thank you for pointing out! We have revised our paper with these citations.
>
> Q6: "we choose a sequential search space (rather than a directed acyclic graph of nodes (Liu et al., 2018b)), i.e., convolutional layers are sequentially stacked in our network": "stacked" is confusing here.
> "Stack" means convolutional layers are sequentially connected, instead of forming up a complex directed acyclic graph like the cells used in DARTS (Liu et al. 2018).
>
> Q7: "we allow each cell to be individually searchable across the whole search space": what do you mean? Anything beyond each cell containing different operators after learning?
> In many cell-based search spaces (Liu et al. 2018; Liu et al. 2019), one searched cell is repeated multiple times to form the entire neural network. In our work, each cell in our network is individually searched without repeat, i.e., they very likely choose different operators and expansion ratios.
>
> Q8: If \alpha = \beta in eq. 6 of appendix C, then w and hence Target(m) does not depend on latency, isn't this a typo?
> We followed the convention in MnasNet paper (Tan et al. 2018) to choose \alpha = \beta = -0.07. We plan to more carefully tune the hyperparameters.
>
> Q9,10: Typos.
> Thank you for pointing out! We have revised our paper and corrected these typos.

---

### Official Review · AnonReviewer2 · 2019-10-24
**Official Blind Review #2**

**Rating:** 6

**Review:**

This paper proposes a neural architecture search (NAS) algorithm which automatically finds a efficient network architecture, FasterSeg, for real time semantic segmentation. In designing a NAS algorithm the author takes cue from recent architectural advances introduced for faster segmentation as well as improved accuracy. For instances a) it explores and integrates multi-resolution branches from BiSeNet during NAS b) simultaneously optimizes the loss for accuracy and latency (as done in CAS algorithm) and c) knowledge distillation for semantic segmentation. However, the usage of these blocks in FasterSeg has been well refined to integrate with NAS search. To be precise, their improved version of latency loss avoids architectural collapse during latency-constrained search and it claims to be the first work to co-search for teacher and student network using NAS. Empirical experiments on benchmark dataset suggests that FasterSeg  is more than 30 percent faster with similar accuracy as state-of-the-art real-time segmentation algorithms.

This paper weakly leans towards rejection. Some of the contributing factors
1) Overall presentation of algorithm leaves one more confused. Perhaps, the paper is targeted at small set of audience who primarily works on NAS. More about specific comment in 'Clarification'.
2) There is not a single concrete contribution. For example, NAS search in semantic segmentation using cells and downsampling rates was done in Auto-Deeplab. Further, resource-constrained search for segmentation was introduced in Zhang et al while distillation for segmentation task was proposed by Liu et al.
3) No doubt that it achieves improved efficiency. But at the cost of accuracy. On Camvid and BDD, the competitive algorithm is 1.7 % better in absolute terms. On cityscapes it performs on par. However, the large improvement in accuracy can be attributed to distillation process (Table 3: absolute 2%), without which the overall performance of NAS is suboptimal.

Clarification:
1. It is not clear what is the form of initial network which is pre-trained for 20 epochs ? My guess is, initial network consists of b=3 branches with L=16 sequential layers and for each cell in a layer, the network pre-trains 5 operators as well as for different expansion ratios. Is it correct ?
2. Can you explain 697 unique paths as well as 10^55 unique combinations ?
3. It is noted that by default b=2 is used. However, in FasterSeg network shown in figure 6, I note three branches s={8,16,32}. Am I missing something ? Also, how more branches will introduce more latency ? Branches operate in parallel with max sensitivity s=0.01 and max L=16.
4. Next, as pointed in 3.4 the discrete architecture is obtained by computing \argmax_l over \beta. In that case, there should only be single connection which branches out from s->2s. In figure 6, I note two branches from 8->16 (4th and 6th cell).
5. If the teacher and student network shares the same weight, then what is the need for distillation ? Only difference I currently note is in the expansion ratio. May be you want to say same pretrained network ?
6. Can you explain with example how \gamma's are updated using backpropagation and lookup-table ?
7. Are you employing STE for Gumbel-Max trick ?
8. The individual terms in eq (3) optimizes for \alpha, \beta and \gamma respectively ?
9. In eq (2), each cell output O is linear combination of different operator ?
10. Once the discrete architecture is obtained, is it retrained on cityscapes from scratch or fine-tuned ?

Request for ablation:
1. What is the variation in NAS output with changes in \lambda ? Precisely, can one tradeoff accuracy for improved latency just by tuning \lambda ?
2. What happens if teacher network is also optimised over \gamma ? The difference between teacher and student will then only be in loss function.
3. Currently, discrete architecture is greedily extracted. This need not be the best. Instead one can utilize sequential beam search (vitterbi algorithm). With this it is possible to visualise the accuracy and latency distribution of, say top 100 architecture obtained by NAS.

Minor comments:
1. Seachable -> Searchable

Updates:
I read through the reviews of other reviewers as well as the rebuttal posted by authors. Overall, I am satisfied with the authors response and hence Improving my scores to Weak Accept.

**Experience Assessment:**

I have published one or two papers in this area.

**Review Assessment: Checking Correctness Of Derivations And Theory:**

N/A

**Review Assessment: Checking Correctness Of Experiments:**

I assessed the sensibility of the experiments.

**Review Assessment: Thoroughness In Paper Reading:**

I read the paper thoroughly.

---

> ### Author Response · Authors · 2019-11-11
> **Response to Review 2 (Ablation studies)**
>
> 1. What is the variation in NAS output with changes in \lambda? Precisely, can one tradeoff accuracy for improved latency just by tuning \lambda?
> We complete ablation experiments on lambda = 0.001 and 0.1. During the search, we can see with lambda = 0.001 the latency is barely minimized, and with 0.1 the latency is quickly decreased. On Cityscapes validation set, lambda of 0.001 gives a model of FPS = 122 and mIoU = 73.2%, while lambda = 0.1 gives a model of FPS = 167 and mIoU = 68.1%. Technically yes, by tuning lambda we can adjust the tradeoff between accuracy and latency.
>
> 2. What happens if teacher network is also optimised over \gamma? The difference between teacher and student will then only be in loss function.
> We complete a search experiment with the teacher also searching for gamma. For expansion ratios in [4, 6, 8, 10, 12], the teacher selects [0%, 3%, 18%, 11%, 68%] respectively. This indicates that the teacher dominantly choose heavy kernels. The core reason is that there is no latency penalty on the teacher, therefore the teacher selects large expansion ratios during the search. As confirmed by reviewer #3, always picking the largest expansion ratios for the teacher is an effective and sensible approach.
>
> 3. Currently, discrete architecture is greedily extracted. This need not be the best. Instead one can utilize sequential beam search (vitterbi algorithm). With this it is possible to visualise the accuracy and latency distribution of, say top 100 architecture obtained by NAS.
> This is a great question. Actually, we carefully analyzed the Viterbi algorithm before, and did not choose it. The core reason is that the transition between layers is not a stationary Markov process, and the layer-wise transition probabilities are imbalanced. For example, as we search for downsample rates s in {8, 16, 32}, cells with s = 8 only have one input from the previous layer, while cells with s = 16 or 32 have two inputs (one from s/2 and one from s from the previous layer). This non-stationary and imbalanced transition will end up with biased sampling in cells with different downsample rates.

---

> ### Author Response · Authors · 2019-11-11
> **Response to Review 2 (Clarifications)**
>
> Q1: It is not clear what is the form of initial network which is pre-trained for 20 epochs.
> The architecture parameters are not updated during the 20-epoch pretraining. All branches / operators / expansion ratios equally contribute to the final output.
>
> Q2: About the 697 unique paths as well as 10^55 unique combinations.
> There actually should be 1695 unique paths and we have revised our paper: sorry for the wrong count in the initial draft.  Since we search downsample rate s in {8, 16, 32} and our L = 16, for the path ends with s = 8, there is only one such path. For paths end with s = 16, there are L - 1 = 15 paths. For paths end with s = 32, there are (L-2) + (L-3) + … + 2 + 1 = 105 paths. As we pick two out of three possible branches, the combination gives us 1*15 + 1*105 + 15*105 = 1695 paths. For each cell, each operator has five expansion ratios except for the skip_connecct, resulting in (1+4*5) variants for each cell. However, since we do not search expansion ratios in the head, i.e., the connections from the last layer to the head have a fixed width. Therefore, (1+4*5) variants are only for cells before the last layer. In sum, we have (1+4*5)^(15+14+13) + 5^3 combinations for cells.
>
> Q3: Number of branches in FasterSeg network shown in Figure 6.
> In Figure 6 there are two branches, one ends up with downsample rate s = 16, the other one ends up with s = 32. The head module also leverages the feature map at 1/8 scale from cell #6, but that is not a searched branch, just a connection. Higher latency from more branches is not due to the sensitivity s=0.01, but from more cells (i.e. more operations) in the new branch. The FPS column of the first and second row in Table 3 can explain this phenomenon.
>
> Q4. Two branches from 8->16 (4th and 6th cell) in Figure 6.
> For s = 32, we need to decide two downsampling positions. To achieve this, we calculate argmax_l (beta^0_s,l * beta^0_2s,l), i.e., the joint probability of two downsampling positions. This joint distribution is different from the marginal distribution, which gives us two different 8->16 downsampling positions. We have revised our paper to make this point clear.
>
> Q5. If the teacher and student network shares the same weight, then what is the need for distillation?
> During the search, the teacher and student share the same supernet weight but there is no distillation. When both the teacher and student are training from scratch, they do not share any weight but there is a distillation loss. The differences from teacher to student during the search are 1) no search on expansion ratio, 2) no latency penalty.
>
> Q6: Can you explain with example how \gamma's are updated using backpropagation and lookup-table?
> Each expansion ratio represents a specific portion of output channels in the previous layer and input channels in the next layer. Using the Gumbel-Max trick, only one expansion ratio is activated during the forward and backpropagation process. Gradients calculated based on the sampled expansion ratios are backpropagated to each categorical distribution. Similarly, in our lookup table, we measure the latency for layers with each expansion ratio in advance, and this latency is accumulated along with the forward process as the latency penalty.
> For example: let’s say for a specific layer the softmax of the gamma_i is [0.15, 0.15, 0.4, 0.15, 0.15], representing the probability of selecting expansion ratios [4, 6, 8, 10, 12]. The one-hot vector sampled by STE Gumbel-Max may be [0, 0, 1, 0, 0]. During forward, we calculate 1*OP(ratio=8), while not activating other ratios. The loss and gradient calculated by using OP(ratio=8) in this layer will be backpropagated to update gamma_i[2]. For the latency, we just switch “OP” by “Latency” which leverages the prepared latency values in our lookup table.
>
> Q7: Are you employing STE for Gumbel-Max trick?
> Yes. We maintain the discrete categorical distribution for expansion ratios and use the Straight-Through (ST) Gumbel Estimator for sampling and backpropagation.
>
> Q8: The individual terms in eq (3) optimizes for \alpha, \beta and \gamma respectively?
> Yes. We only calculate gradients for \alpha, \beta, and \gamma respectively in each individual terms in eq (3).
>
> Q9: In eq (2), each cell output O is linear combination of different operator?
> Yes, that is correct.
>
> Q10: Once the discrete architecture is obtained, is it retrained on cityscapes from scratch or fine-tuned ?
> Each time when an architecture is obtained after the search, it will be trained from scratch on the dataset, without inheriting or fine-tuning from the weights used in the supernet.

---

> ### Author Response · Authors · 2019-11-11
> **Response to Review 2**
>
> Thank you for your time and comments. We have revised our paper and we believe our responses and revisions address all your concerns. We would be very grateful if you would please look over our paper again, and consider changing your scores.
>
> Indeed, our paper primarily targets at contributions to the NAS fields.  However, we also expect the work to provide broad reference values to the general audience working on related computer vision problems. We apologize if our paper didn’t read smoothly to you at the first glance: we have addressed all your specified “clarification” comments and hope our work’s merit now becomes more clear to you.
>
> We respectfully cannot agree on your comment “there is not a single concrete contribution”. We apologize if your misunderstanding arose from our lack of writeup clarity or else. Please see our detailed explanations below.
>
> First of all, we are well aware of NAS search works done in semantic segmentation. Meanwhile, in most NAS papers published, designing novel search spaces and improving the search algorithms are considered as the two core contributions to claim: see [1,2,3] for example. They are also what makes FasterSeg substantially different and novel from Auto-DeepLab.  As confirmed by other reviewers, both our multiscale search space, the regularized latency optimization, and the co-searching algorithm are significant contributions. They are for the first time proposed, well-motivated, and supported by our solid ablation studies. Furthermore, we push the performance of real-time segmentation to a new state of the art.
>
> Second, our distillation is not appended as a post-processing to the searched/designed models as done in the mentioned literature. Instead, integrating distillation into the NAS framework (and therefore enabling jointly search multiple networks) is the KEY. This is confirmed by the last two rows in Table 3, where we distill to the pruned teacher net and to our search student net respectively, from the same teacher network. With the same distillation training setting, the pruned teacher is 6% worse than our searched student, indicating that the important reason for our FasterSeg’s superior performance is in the optimized architecture.
>
> In sum, while we all agree that neither “(efficient) NAS for segmentation” nor “distillation” is novel now, our methods’ contributions are concretely established, well beyond those. Hopefully, the above answers have resolved your confusion on our novelty.
>
> In the experiments, by further fine-tuning our FasterSeg, we are now able to achieve 71.5% on Cityscapes test set, bypassing all previous works on both mIoU and FPS in Table 4, making CityScape our clear all-win case. In Table 5 and 6, our FasterSeg now achieves 71.1% and 54.9% respectively, reaching the same bar on mIoU as CAS and DRN, while surpassing their latency by over-doubling the FPS.
>
> Notice that, we target at extremely efficient segmentation, for our own specific application needs at very high FPSs (~150 - ~400). Such is considered as "well-motivated" by the other two reviewers. To meet that demanding requirement, our model is clearly the best option available that can still maintain state-of-the-art mIOUs in addition to its unparalleled efficiency.
>
> We note that the other two reviewers concur and appreciate our paper’s novelty points well. We are more than happy to provide extra clarification and justifications if needed.
>
> [1] Bender, Gabriel, Pieter-Jan Kindermans, Barret Zoph, Vijay Vasudevan, and Quoc Le. "Understanding and simplifying one-shot architecture search." ICML 2018.
> [2] Liu, Chenxi, Barret Zoph, Maxim Neumann, Jonathon Shlens, Wei Hua, Li-Jia Li, Li Fei-Fei, Alan Yuille, Jonathan Huang, and Kevin Murphy. "Progressive neural architecture search." ECCV 2018.
> [3] Cai Han, Ligeng Zhu, and Song Han. "Proxylessnas: Direct neural architecture search on target task and hardware." ICLR 2019.

---

### Decision · Program_Chairs · 2019-12-19

**Decision:**

Accept (Poster)

**Comment:**

This paper presents neural architecture search for semantic segmentation, with search space that integrates multi-resolution branches. The method also uses a regularization to overcome the issue of learned networks collapsing to low-latency but poor accuracy models. Another interesting contribution is a collaborative search procedure to simultaneously search for student and teacher networks in a single run. All reviewers agree that the proposed method is well-motivated and shows promising empirical results. Author response satisfactorily addressed most of the points raised by the reviewers. I recommend acceptance.